# Does Advanced Maternal Age Comprise an Independent Risk Factor for Caesarean Section? A Population-Wide Study

**DOI:** 10.3390/ijerph20010668

**Published:** 2022-12-30

**Authors:** Anna Šťastná, Tomáš Fait, Jiřina Kocourková, Eva Waldaufová

**Affiliations:** 1Department of Demography and Geodemography, Faculty of Science, Charles University, 128 00 Praha, Czech Republic; 2Department of Gynaecology and Obstetrics, 2nd Faculty of Medicine, Charles University, 150 06 Praha, Czech Republic; 3Department of Health Care Studies, College of Polytechnics Jihlava, 586 01 Jihlava, Czech Republic

**Keywords:** caesarean section (CS), maternal age, fertility postponement, marital status, education, Czechia

## Abstract

Objective: To investigate the association between a mother’s age and the risk of caesarean section (CS) when controlling for health factors and selected sociodemographic characteristics. Methods: Binary logistic regression models for all women who gave birth in Czechia in 2018 (N = 111,749 mothers who gave birth to 113,234 children). Results: An increase in the age of a mother significantly increases the odds of a CS birth according to all of the models; depending on the model, OR: 1.62 (95% CI 1.54–1.71) to 1.84 (95% CI 1.70–1.99) for age group 35–39 and OR: 2.83 (95% CI 2.60–3.08) to 3.71 (95% CI 3.23–4.27) for age group 40+ compared to age group 25–29. This strong association between the age of a mother and the risk of CS is further reinforced for primiparas (probability of a CS: 11% for age category ≤ 19, 23% for age category 35–39, and 38% for age category 40+). However, the increasing educational attainment of young women appears to have weakened the influence of increasing maternal age on the overall share of CS births; depending on the model, OR: 0.86 (95% CI 0.80–0.91) to 0.87 (95% CI 0.83–0.91) for tertiary-educated compared to secondary-educated women. Conclusions: The age of a mother comprises an independent risk factor for a CS birth when the influence of health, socioeconomic, and demographic characteristics is considered.

## 1. Introduction

A fertility shift to higher ages over the last few decades comprises one of the most distinctive features of reproductive behavior in developed countries [1,2]. Many studies have linked an increasing maternal age with adverse pregnancy outcomes [3], higher risks of preterm birth and low birth weight [4,5], stillbirth and unexplained fetal death [6,7,8], and an increasing proportion of caesarean section births [3,9,10].

Increases in caesarean section (CS) rates have been observed globally over the past few decades, and almost doubled between 2000 and 2015: from 12% in 2000 to 21% of births in 2015 [11].

The same trends concerning fertility postponement and an increase in CS rates are evident in Czechia, where the fertility postponement process started as late as the 1990s and has been particularly dynamic [2]; the mean age of women at childbirth increased from 24.8 in 1990 to 30.1 in 2018, and the percentage of live births to mothers aged 35 and over rose from just 4% in 1990 to 22% in 2018 [12]. The CS rate more than doubled in this period: from 10% in 1994 to 24% in 2018 (latest available data) [13].

The significant increase in maternal age has been accompanied by major structural changes—the educational level of younger women has risen significantly and forms of relationship have changed from the married state to cohabitation, singlehood, single motherhood, and re-partnering. Moreover, women at older ages are giving birth to children of lower parity [14]. The use of assisted reproductive techniques has increased significantly in Czechia over the last two decades and has contributed to the increase in multiple pregnancies since the second half of the 1990s [15,16].

Some international studies have suggested that better socioeconomic conditions mitigate some of the adverse effects of advanced maternal age on perinatal outcomes [7,17], while studies on the link between CS and maternal social background have reported conflicting results [3,18,19].

Due to continuous fertility postponement [2], research on the effect of advanced maternal age on the risk of CS is becoming increasingly important due to its contribution to determining strategies aimed at reducing CS rates. However, detailed data are lacking on the various factors that contribute to high CS rates. Although a study has been conducted on the link between advanced maternal age and CS [9], it lacked important information on the use of ART, which is also associated with increased maternal age.

The aim of this study is to estimate the association between a mother’s age and the risk of CS while adjusting for other factors—sociodemographic and health characteristics, including ART use, that may be related to advanced maternal age. The following analysis employs binary logistic regression models and a unique data source containing anonymized information on all mothers in Czechia taken from the National Register of Reproduction Health database for 2018, which allows for the linkage of selected sociodemographic and health characteristics.

## 2. Methods

### 2.1. Data Sources

This analysis is based on statistical evidence of births recorded by the Institute of Health Information and Statistics of the Czech Republic (IHIS) and on anonymized individual data on mothers who gave birth in Czechia in 2018 obtained from the National Register of Reproduction Health, module of mothers (IHIS). The register covers the whole population; the collection of such medical data is required by legislation and must be provided by the health facility at which the delivery or postnatal care of a mother and newborn occurred. The module of mothers contains data on the reproductive history of mothers, the courses of pregnancies, childbirths, and new-born children. The datasets were supplemented with information from the Assisted Reproduction module that monitors assisted reproduction technique (ART) treatment in Czechia, thus creating a unique opportunity for distinguishing those mothers who became pregnant and subsequently gave birth following the use of ART.

The analytical sample comprised 111,749 mothers who gave birth to 113,234 children in 2018 in Czechia. Since 21% of the records did not indicate the woman’s level of education and/or marital status, we used a subsample of 88,041 mothers for the partial analysis of the effects of education and marital status. The subsample did not differ from the total population in terms of the frequency of CS, the age composition, or the structure of the women across all of the variables included in the models (the highest measured difference in the share of women per variable between the two samples did not exceed 0.01 percentage points). Thus, the reduction in the sample size did not bias the analysis of the women’s education and marital status.

### 2.2. Analytical Methods

We employed binary logistic regression in order to identify the various aspects of the associations between the age of mothers at childbirth, CS, and the influence of sociodemographic, health, and obstetric factors.

The dependent variable in the binary logistic regression was set at 1 for a CS (including both planned and emergency CS) and 0 for a vaginal delivery. The model equation was as follows:(1)$$logit\left(\mathrm{Pr}\langle Y=1|x\rangle \right)=\log\left\{\frac{\mathrm{Pr}\langle Y=1|x\rangle }{1-\mathrm{Pr}\langle Y=1|x\rangle }\right\}=\beta_{0}+\beta_{1}x_{1}+\ldots +\beta_{k}x_{k},\;$$
where *Y* is the dependent variable (*Y* = 1 for a CS, otherwise *Y* = 0), *x* = (*x*_1_, …. *x_k_*)’ is the explanatory variables vector, *β*_0_ is the intercept parameter, and *β* is the slope parameter vector.

For the sake of clarity, the results were interpreted in terms of odds:(2)$$\begin{array}{cc} \frac{\mathrm{Pr}\langle Y=1|x\rangle }{1-\mathrm{Pr}\langle Y=1|x\rangle } & =\exp\left[logit\left(\mathrm{Pr}\langle Y=1|x\rangle \right)\right]=\exp\left(\beta_{0}+\beta_{1}x_{1}+\beta_{k}x_{k}\right)\\ & =\exp\left(\beta_{0}\right)\times \exp\left(\beta_{1}x_{1}\right)\times \ldots \times \exp\left(\beta_{k}x_{k}\right), \end{array}$$

The odds ratios (Exp(*β*) in the table) indicate the odds of a CS for a given category compared to the reference category, while controlling for the other covariates.

We also calculated the probability of a CS delivery for the defined groups of women based on the regression coefficients, as estimated by the resulting model:(3)$$Pr\left(Y=1\right)=\frac{\exp\left(\beta_{0}+\sum \beta_{k}X_{k}\right)}{1+\exp\left(\beta_{0}+\sum \beta_{k}X_{k}\right)}$$

We constructed five binary logistic regression models—Models 1 and 2 for all of the mothers in addition to Model 3 for all of the mothers with information on their highest educational attainment and marital status. In line with the literature [9,19], we then focused specifically on primiparas (Model 4) and on the obstetrically low-risk group of mothers (women with singleton pregnancies who experienced no complications in pregnancy and childbirth, no diabetes, no CS for a previous delivery, and no breech position, Model 5) to observe the effect of age with regard to a more homogeneous group unconfined by the main clinically documented factors that influence the risk of a CS. Given the large sample sizes, the models were robust. A number of maternal sociodemographic, health, and pregnancy characteristics were included in the models as explanatory variables:

The main explanatory variable was the age of a mother at the time of delivery, which was categorized into six age groups: ≤19 years, 20–24 years, 25–29 years (ref.), 30–34 years, 35–39 years, and 40 years and above.

The other maternal sociodemographic characteristics comprised marital status (single—married (ref.)—divorced—widowed) and the highest attained level of education (basic (including incomplete)—secondary without a school leaving certificate (SLC)—secondary with an SLC (ref.)—tertiary).

The pregnancy characteristics included gestational age (following the WHO [20] classification and Spong [21], we applied 5 categories: extremely preterm (<28 weeks)—very preterm (28 to <32 weeks)—moderate to late preterm (32 to <37 weeks)—37–41 weeks as the reference category—post-term births (42+ weeks)), parity (first (ref.)—second—third and higher birth orders), and singleton/multiple gestation (singleton (ref.) vs. multiple pregnancy). The incidence of breech position was also controlled in the models (no (ref.) vs. yes). In the case of multiple pregnancies, a pregnancy was classified as “yes” if at least one of the children was in the breech position.

Maternal health characteristics included the probable method of pregnancy estimated according to the embryo transfer date reported in the register of assisted reproduction (without the use of ART (ref.) vs. following ART), the incidence of diabetes (not detected (ref.)—detected prior to pregnancy—detected during pregnancy), health complications in pregnancy (no complications (ref.)—hypertension—threatened preterm labor—other complications category, including bleeding in the first, second, and third trimesters, placenta previa, placental abruption and other placental abnormalities, cardiovascular complications, pre-eclampsia, and intra-uterine growth restriction), and previous CS delivery (no (ref.) vs. yes).

Since the multi-collinearity test revealed high multi-collinearity between birth weight and gestational age, this variable was excluded from the final models.

The models did not include variables related to the perinatal care provided to mothers due to the conditions of the obstetric system in Czechia, where the care of pregnant women is fully entrusted to gynecologists and obstetricians. Czech legislation does not permit home births since midwives are not allowed to work outside hospital facilities [22]. In the event that a birth in a medical facility is conducted by a midwife, the doctor remains the legally responsible person. The latest available data indicate that 80.2% of births were managed by doctors and 19.7% by midwives in 2015 [23]. Despite some maternity facilities being run by private companies, all healthcare is covered by the public health insurance system under the same conditions, i.e., regardless of the type of healthcare facility. Health insurance is compulsory for all residents of Czechia. Caesarean sections are either planned during pregnancy or are conducted in response to events that arise during labor. No option exists for a caesarean section on request; however, in recent years the woman’s opinion has been taken into account to an increasing extent, e.g., in the case of the breech position or the condition following a caesarean section. Indications are emerging that essentially act to legalize caesarean sections on request, e.g., protection of the pelvic floor [24,25].

## 3. Results

The increase in the share of CS births in Czechia occurred with dynamics that differed with respect to both calendar years and age groups. Figure 1 shows that the increase was moderate during the 1990s and accelerated significantly after 2000. The largest increase in the share of CS births during the observed period related to the 20–24 age group, for whom the proportion of births more than doubled between 1994 and 2018 (Figure 1). A smaller increase was observed for women aged 25–29 (twofold increase) and over 30, while the proportion of CS births in the 30–34 age group increased by 1.8 times between 1994 and 2018, and by 1.5 times for women over 35. In general, the proportion of CS births increased with the increasing age of a mother, regardless of the year. In 2018, the proportion of CS births to women over 35 years of age was more than 30%, compared to less than 20% to women below 25 years of age (Figure 1).

The research results revealed the effect of age on the odds of a CS adjusted for the various independent covariates according to several models that progressively homogenized the groups of women studied. Table 1 shows that the odds of a CS birth increase significantly with the increasing ages of mothers after controlling for the above-mentioned pregnancy and maternal health characteristics (Model 1). After adding the interaction between the mother’s age and the birth order (Model 2), the effect of age is even stronger for women over 40 years of age, i.e., the odds of a CS birth for this age group is 3.6 (95% CI 3.15–4.04) times higher than that of the 25–29 age group.

A more detailed analysis of age combined with parity shows that the effect of maternal age reflects differing dynamics according to the birth order (Figure 2). Primiparas in all of the age groups have a higher probability of giving birth via CS, which, moreover, increases significantly with age. Meanwhile, at the age of 25–29 years, which is the most common age for childbirth in Czechia, the probability of a CS is 15%; it is 23% for those aged 35–39 and for primiparas over 40 years it is close to 40% (Figure 2) when controlling for other variables, including health complications (Model 2). With respect to secundipara, the probability of giving birth via CS is around 5% up to the age of 35, which increases slightly after 35 years and almost triples to 14% at 40 years of age plus. Concerning third and higher-order births, the probability of a CS birth is below 5% up to age 35 and increases slightly to 8% for women 40 years of age plus (Figure 2).

We extended the analysis to include the control of selected sociodemographic characteristics of the mothers—the highest educational attainment and marital status (Model 3). Moreover, we also focused on two subgroups of mothers, aiming to monitor the effect of age on the risk of a CS birth in specific groups and to eliminate certain potential effects that were previously only controlled for in the complete models. Table 2 presents the effect of age for the following groups: (a) primiparas (Model 4) and (b) the obstetrically low-risk group (Model 5), which includes only women with singleton pregnancies who had no complications in pregnancy and childbirth, no diabetes, no CS for a previous delivery, and no breech position. In addition to age, the influence of the other relevant explanatory variables specified in the Methods section was controlled for in each of the models.

Table 2 shows that, after controlling for sociodemographic characteristics, the effect of increasing age on the increasing odds of a CS birth remains significant. In the case of primiparas (Model 4), the effect of age is, again, more pronounced: the youngest primiparas under the age of 20 are around half as likely to have a CS birth as those aged 25–29, whereas the odds of a CS increase by 3.71 (95% CI 3.23–4.27) times for primiparas aged over 40 years compared to the 25–29 age group. We obtained similar results for the obstetrically low-risk group, i.e., singleton pregnancies where no major health CS indications were detected. Here too the odds of a CS birth increase with age, and are 1.83 (95% CI 1.69–1.98) times higher for the 35–39 age group and 3.58 (95% CI 3.15–4.06) times higher for mothers aged 40+ compared to the 25–29 age group.

Table 2 shows that higher educational attainment reduces the odds of a CS birth compared to those with a secondary education with an SLC according to all three models. Conversely, concerning the obstetrically low-risk group, the odds of a CS birth are higher for those in the lowest education group.

Marital status differentiates the odds of a CS birth in the case of divorced and single women—both groups have higher odds of a CS birth compared to married women, even in the case of the obstetrically low-risk group.

## 4. Discussion

### 4.1. Main Findings

This study aimed to determine the effect of a mother’s age at birth on the risk of CS in the context of intensive fertility postponement and significant structural changes regarding mothers’ sociodemographic and partner conditions. We revealed the effect of age as adjusted for certain health complications associated with advanced maternal age. The results revealed a strong positive association between maternal age and the odds of a CS birth, which was particularly strong for primiparas and the obstetrically low-risk group after controlling for other (health and sociodemographic) variables. Primiparas have significantly higher odds of a CS birth, which increases dramatically with increasing age—OR: 1.29 (95% CI 1.21–1.38) for age group 30–34, OR: 1.84 (95% CI 1.70–1.99) for age group 35–39, and OR: 3.71 (95% CI 3.23–4.27) for age 40+ compared to primiparas 25–29 years of age. The effect of age on the higher odds of a CS birth in the obstetrically low-risk group was similar: OR: 1.28 (95% CI 1.20–1.37) for age group 30–34, OR: 1.83 (95% CI 1.69–1.98) for age group 35–39 and OR: 3.58 (95% CI 3.15–4.06) for age 40+ compared to the 25–29 age group.

### 4.2. Strengths and Limitations

The main strength of our study concerns the use of data on the whole population of women who gave birth in 2018 in Czechia, thus rendering selection bias unlikely. A further advantage of the data lay in the fact that we could control for several groups of intervening variables—health indications such as hypertension and diabetes, a previous CS birth, multiple pregnancies, parity, and sociodemographic characteristics (the education and marital status of the mother), including the use of ART, which correlates with increasing maternal age.

Despite the comprehensive dataset, the study has several limitations—information on education and marital status is not available for all of the women (dataset reduction of 21%); nevertheless, no selection bias was observed regarding the structure of the mothers by age, CS births, or the other variables included in the models. Moreover, information on ART use was estimated based on information on ART cycles performed in Czechia only; since Czechia is more likely to be a destination country for cross-border reproductive care [26], we did not anticipate any bias in this covariate. The data contain no other sociodemographic characteristics that would allow for further analysis. Similarly, since we had no information on maternal pre-pregnancy weight and height, we were unable to adjust for body mass index (BMI). As in other developed countries, the Czech adult population has a tendency towards overweightness, and the proportion of obese persons (BMI >= 30) has increased substantially over recent decades [27]. Moreover, clinical studies suggest that a higher BMI at the start of a pregnancy significantly increases the risk of a caesarian delivery [28]. Therefore, we considered it essential to include the height and weight information provided in the statistical reports on obstetric care. Such information allows for a detailed analysis of the effect of BMI in population-wide studies.

### 4.3. Interpretation

Fertility postponement has remained a distinctive feature of reproduction patterns in developed countries for several decades, including Czechia, where a particularly dynamic fertility postponement process commenced in the 1990s: the CS rate more than doubled (from 10.3% in 1994 to 23.6% in 2018) and a significant increase in the proportion of multiple births was recorded, from 9.6 multiple births per 1000 births in 1994 to a peak of 21 multiple births per 1000 births in 2010. Even though the multiple birth rate subsequently declined to 13 per 1000 births (2018), due most likely to a change in ART legislation that promoted single-embryo over multiple-embryo transfers, the multiple birth rate stabilized at a higher level than it was in the pre-fertility postponement era.

Our results are consistent with several studies that highlight the significant impact of the maternal age on CS births [9,29,30]. The increase in the total share of CS births in Czechia in the observed period was due to (1) the expansion of CS at all ages and (2) a change in the age structure of mothers. Due to fertility postponement, the share of mothers giving birth at older ages has increased significantly, accounting for 22% of the overall increase in the CS rate between 1994 and 2018 [13]. Moreover, given that older primiparas have the highest probability of a CS birth, the increase in CS rates can also be attributed to a marked change in the structure of women in regard to both age and parity [14]. In addition to advanced maternal age, it was confirmed that the increase in the CS rate in Czechia is also related to the increase in ART use and multiple pregnancies [25]. Moreover, changes in the attitudes of healthcare providers have played a role in this respect, particularly the promotion of elective repeat CS and a move away from spontaneous delivery when the fetus is in the breech position.

Conversely, it seems that a significant increase in the share of tertiary-educated women in Czechia over the last 30 years has weakened the influence of increasing maternal age on the share of CS births. According to census data, the proportion of tertiary-educated women in the population, aged 15+, increased almost fourfold between 1991 and 2021; a similar increase is evident for women of reproductive age, i.e., 40% of the 25–34 years old group were university graduates in 2017 [31]. While increasing maternal age significantly increases the odds of a CS birth, a higher education level was shown by all three models to reduce the odds by 13–14% compared to secondary-educated mothers (including primiparas and the obstetrically low-risk group). Similarly, a higher incidence of CS births for the least educated group of women and a decrease in CS for highly educated women have been demonstrated in Norway, which were explained by both the increasing medical vulnerability of the lowest-educated women in society, in which the overall education level is rising, and a possible increase in CS on request concerning the least-educated group of women [19:847]; this factor, however, is not officially monitored in Czechia. Moreover, the increasing promotion of the benefits of natural childbirth over the last two decades has probably also played a role, a trend that is more likely, at least initially, to be followed by more highly educated expectant parents, as shown for South Korea [32].

Health complications before and during pregnancy comprise a common explanation for the increased risk of a CS birth for older mothers, since the risk of many related diseases increases with maternal age [33]. Our data enabled us to control for several characteristics that are closely related to a higher risk of a CS birth. However, even after controlling for these variables (via the general models or in the models that considered selected subgroups), a significant association remained between increasing maternal age and the increasing odds of a CS birth. Concerning the more homogeneous subgroups, this association was even stronger, and the odds of a CS birth at advanced ages was higher.

## 5. Conclusions

Using a dataset of all births in Czechia in 2018, we demonstrated an increase in the risk of a CS birth with increasing maternal age when controlling for the influence of other covariates (health and sociodemographic). Applying binary logistic regression, we proved that advanced maternal age is an independent risk factor for a CS birth. The effect of age increases with declining parity. We showed that delaying reproduction increases the risk of a CS birth, even for women considered obstetrically low-risk due to their having had a singleton pregnancy without any associated health complications.

The results support the necessity for the improved awareness of future parents of the potential risks of delaying fertility regarding both CS and the various negative health impacts on mothers and children. The results also point to the potential for further research into the factors that are not captured in the statistical databases but that may also act to increase the risk of CS at older ages, such as the fears of mothers and obstetricians, social norms, and the approaching end of the reproductive period.

## Figures and Tables

**Figure 1 ijerph-20-00668-f001:**
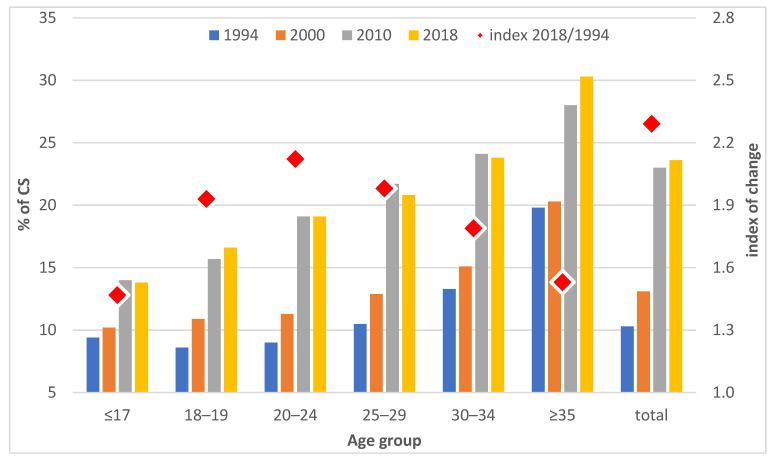
Share of CS births (in %) and index of the change in the share of CS births between 1994 and 2018, Czechia. Source: IHIS 2000, 2015, 2018. Authors’ own calculations.

**Figure 2 ijerph-20-00668-f002:**
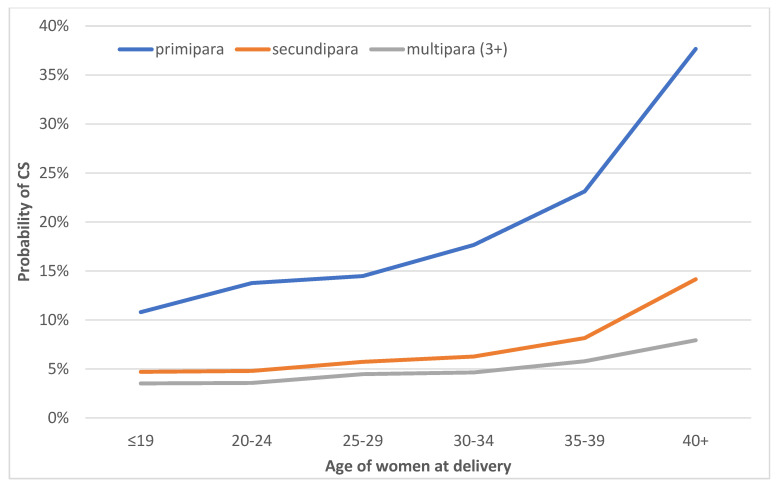
Probability of a CS by maternal age and parity, all deliveries, Czechia, 2018. Based on the results of Model 2, which was adjusted for gestational age, singleton/multiple deliveries, previous CS delivery, ART usage, the incidence of diabetes, health complications in pregnancy, and breech position.

**Table 1 ijerph-20-00668-t001:** Odds ratios (OR) with a 95% confidence interval (CI) for a CS. All deliveries, Czechia, 2018.

	Model 1			Model 2		
	B	Exp(B) 95% CI	*p*-Value	B	Exp(B) 95% CI	*p*-Value
Age of mother						
≤19	−0.333	0.72 (0.63–0.81)	0.000	−0.335	0.71 (0.62–0.82)	0.000
20–24	−0.100	0.90 (0.85–0.96)	0.001	−0.059	0.94 (0.88–1.01)	0.098
25–29		1			1	
30–34	0.188	1.21 (1.16–1.26)	0.000	0.236	1.27 (1.20–1.34)	0.000
35–39	0.485	1.62 (1.54–1.71)	0.000	0.575	1.78 (1.66–1.91)	0.000
≥40	1.041	2.83 (2.60–3.08)	0.000	1.272	3.57 (3.15–4.04)	0.000
N	111,749					

Note: Model 1 was adjusted for gestational age, parity, singleton/multiple deliveries, previous CS delivery, ART usage, the incidence of diabetes, health complications in pregnancy, and breech position. Model 2 was also adjusted for the interaction between the mother’s age and parity. The models are shown in full in the supporting information provided in Appendix A.

**Table 2 ijerph-20-00668-t002:** Odds ratios (OR) with a 95% confidence interval (CI) for a CS; the models that control for sociodemographic characteristics and models for selected groups (primipara and the obstetrically low-risk group), Czechia, 2018.

	Model 3			Model 4—Primipara			Model 5—Obstetrically Low-Risk Group		
	B	Exp(B) 95% CI	*p*-Value	B	Exp(B) 95% CI	*p*-Value	B	Exp(B) 95% CI	*p*-Value
**Age of mother**									
≤19	−0.463	0.63 (0.54–0.74)	0.000	−0.556	0.57 (0.48–0.69)	0.000	−0.743	0.48 (0.38–0.59)	0.000
20–24	−0.132	0.88 (0.82–0.94)	0.000	−0.117	0.89 (0.82–0.97)	0.006	−0.172	0.84 (0.77–0.93)	0.000
25–29		1			1			1	
30–34	0.192	1.21 (1.15–1.27)	0.000	0.256	1.29 (1.21–1.38)	0.000	0.248	1.28 (1.20–1.37)	0.000
35–39	0.483	1.62 (1.53–1.72)	0.000	0.608	1.84 (1.70–1.99)	0.000	0.603	1.83 (1.69–1.98)	0.000
≥40	1.059	2.88 (2.62–3.18)	0.000	1.312	3.71 (3.23–4.27)	0.000	1.274	3.58 (3.15–4.06)	0.000
**Education**									
Basic	−0.006	0.99 (0.89–1.08)	0.886	0.112	1.12 (1.00–1.25)	0.052	0.160	1.17 (1.05–1.31)	0.006
Secondary without SLC	−0.031	0.97 (0.92–1.02)	0.260	−0.019	0.98 (0.92–1.05)	0.587	0.003	1.00 (0.93–1.08)	0.931
Secondary with SLC		1			1			1	
Tertiary	−0.138	0.87 (0.83–0.91)	0.000	−0.139	0.87 (0.82–0.93)	0.000	−0.156	0.86 (0.80–0.91)	0.000
**Marital status**									
Single	0.054	1.06 (1.01–1.10)	0.011	0.065	1.07 (1.01–1.13)	0.017	0.061	1.06 (1.01–1.12)	0.031
Married		1			1			1	
Divorced	0.152	1.16 (1.06–1.28)	0.002	0.057	1.06 (0.91–1.24)	0.470	0.245	1.28 (1.12–1.46)	0.000
Widowed	−0.150	0.86 (0.50–1.48)	0.588	0.160	1.17 (0.50–2.78)	0.716	0.086	1.09 (0.49–2.40)	0.831
N	88,041			41,914			59,424		

Note: Model 3 was adjusted for gestational age, parity, singleton/multiple delivery, previous CS delivery, ART usage, the incidence of diabetes, health complications in pregnancy, and breech position. Model 4 (primipara) was adjusted for gestational age, singleton/multiple delivery, ART usage, the incidence of diabetes, health complications in pregnancy, and breech position. Model 5 (obstetrically low-risk group) was adjusted for gestational age, parity, and ART usage. The models are shown in full in the supporting information provided in Appendix A.

## Data Availability

IHIS—Institute of Health Information and Statistics of the Czech Republic. Výsledky ze zpracování “Zpráva o novorozenci” za roky 1994–1996 (Results from the processing of the “Newborns Report” for 1994–1996). 2000. Prague: The Institute of Health Information and Statistics of the Czech Republic; 2000; IHIS—Institute of Health Information and Statistics of the Czech Republic. Rodička a novorozenec 1999–2015 (Mothers and newborns 1999–2015). The Institute of Health Information and Statistics of the Czech Republic; 2015. Available online: https://www.uzis.cz/index.php?pg=vystupy--knihovna&id=249 (accessed on 27 December 2022); IHIS—Institute of Health Information and Statistics of the Czech Republic. Anonymized individual data from the National Register of Reproduction Health—Module on Mothers linked with data on ART from the NRRH Assisted Reproduction module. 2018. Data collected on request for scientific purposes only, not publicly available.

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
