# Peer review of "Does Advanced Maternal Age Comprise an Independent Risk Factor for Caesarean Section? A Population-Wide Study"

_ijerph, 2022, doi:10.3390/ijerph20010668_

Round 1
Reviewer 1 Report
Thank you for the privilege of reviewing your research. The research project aimed to estimate the association between the mother’s age and the risk of CS while adjusting for other factors – socio-demographic and health characteristics including ART use that may be related to advanced maternal age. The article is a valuable resource on the subject.
Due to continuous fertility postponement, research on the effect of advanced maternal age on the risk of CS is becoming increasingly important due to its contribution to determining strategies aimed at reducing CS rates.
It would also be useful to show for what reason caesarean sections were performed. The frequency of pregnancy complications also increases with age, which affects the risk of caesarean sections.
Author Response
Thank you for your suggestions to improve the article. Concerning the health indications for the caesarean section. In general, the information on the indication for CS is tracked in the data. However, this is not a mandatory item in the statistical report, therefore this information is missing for the majority of CS deliveries (specifically, out of 26,000 caesarean section deliveries, we have listed medical indications for CS in just under 7.7 thousand deliveries). Thus, we do not include a more detailed analysis in our text, because of the risk of bias in the results (it is not clear why this data is not filled in for a larger proportion of births; it may also be a systematic practice of some health facilities). However, we agree that this is a very interesting topic and there is scope for further detailed investigation including specific medical indications for CS if the statistical recording is improved.
We agree that the frequency of pregnancy complications also increases with age, which affects the risk of caesarean sections. Therefore, in the models, in addition to the age of the mother and other socio-demographic characteristics, we also control for health complications identified before or during pregnancy. Alternatively, in some models, we include women without any of the observed health complications.
Reviewer 2 Report
Interesting manuscript.
Comments:
- The "big unmentioned" is weight/BMI, which correlates with age. BMI is reckoned to be a crucial explanatory factor for the rising CS rate. BMI increases up to about the seventh decade. I understand that the authors have no data on this, it is less understandable that they simply ignore it.
- Methods, "Czech specifics". "Czech legislation does not permit childbirth at home". That is, unless it happens by nature, in the toilet, or in the car, I guess even Czech women are not immune to the sometimes frightening event of delivering outside of the safety of a hospital bed. And I doubt that the ban to deliver at home is 100% loophole-clear.
"The decision on a planned CS cannot simply be based on a request". I doubt very much that this is happening. Women with an anxiety-disorder all over the world are in some (yes, perhaps rare or very rare) cases granted a primary CS, in some countries this even counts as a specific indication: "tokophobia". I guess Czech doctors can simply dust this under the carpet as a "psychiatric" indication, or a variant of this. You cannot force all medically eligible women to go through labour.
- "Marital status" (table 2) yields some interesting data. The risk of CS is higher in single or "divorced" women. Apart from the antiquated social stratification, one wonders what factors may be underneath: again BMI? substance use (smoking, drugs), which may lead to pathological pregnancies and labour? Do the authors have data on substance use?
Author Response
First of all, we thank you for your detailed review of the manuscript and for your suggestions on how to improve the manuscript. We have incorporated them into the manuscript.
We have modified the wording of two sentences to be more precise in describing the conditions of the system:
Instead of the sentence “Czech legislation does not permit childbirth at home ....“
new formulation: „Czech legislation does not permit home births since midwives are not allowed to work outside hospital facilities [22].“
Instead of the sentence "The decision on a planned CS cannot simply be based on a request".
new formulation: „There is no option for a caesarean section on request. However, in recent decades, the woman's opinion has been taken into account considerably, e.g. in the case of the breech position or the condition after a caesarean section. Indications are emerging that essentially legalize caesarean sections on request, e.g. protection of the pelvic floor [24,25].“
We added information on BMI, the effect of BMI on CS risk, and implications for future improvements in statistical reporting by including BMI information.
Concerning marital status - It is important to note that in the Czech Republic, half of all children are born to unmarried women (in 2018, 48.5% of children were born out of wedlock, and 57% of parity 1 children). While it is true that out-of-wedlock fertility is increasing along with the decreasing educational level of women, it is otherwise not a selective group of women with significantly riskier behaviour in terms of, for example, substance abuse or BMI. We control for education in the models as well, so education should not be hidden behind the higher risk of CS for unmarried women. Conversely, we do not have information on substance abuse in pregnancy and BMI in the dataset, but we do not expect a significant effect here simply because out-of-wedlock fertility affects half of all births. The influence of marital status would undoubtedly merit deeper analysis, but that is beyond the scope of the research question addressed here.
Reviewer 3 Report
The study is well described and the conclusions are mostly clear.
I would like to ask if the authors can give some suggestions (page 8, line 284) which could be the other causes beyond maternal age that can be responsible of the increase in CS rate.
I prefer Caesarean section because the word comes from Julius Caesar)!
Author Response
Other causes beyond maternal age that can be responsible for the increase in CS rate in Czechia were added together with the reference to another study that was focused on the examination of all factors associated with a CS birth.
After consulting the literature, we retain "caesarean section" instead of "Caesarean section" because both options could be found in publications. However, if the editors request a modification to capital C, we are not against it.